# STORM: Efficient Stochastic Transformer based World Models for Reinforcement Learning

**Weipu Zhang, Gang Wang,** [*] **Jian Sun, Yetian Yuan**
National Key Lab of Autonomous Intelligent Unmanned Systems, Beijing Institute of Technology
Beijing Institute of Technology Chongqing Innovation Center
zhangwp.bit@gmail.com, {gangwang,sunjian,ytyuan}@bit.edu.cn

**Gao Huang**
Department of Automation, BNRist, Tsinghua University
gaohuang@tsinghua.edu.cn

## Abstract

Recently, model-based reinforcement learning algorithms have demonstrated remarkable efficacy in visual input environments. These approaches begin by constructing a parameterized simulation world model of the real environment through self-supervised learning. By leveraging the imagination of the world model, the agent's policy is enhanced without the constraints of sampling from the real environment. The performance of these algorithms heavily relies on the sequence modeling and generation capabilities of the world model. However, constructing a perfectly accurate model of a complex unknown environment is nearly impossible. Discrepancies between the model and reality may cause the agent to pursue virtual goals, resulting in subpar performance in the real environment. Introducing random noise into model-based reinforcement learning has been proven beneficial. In this work, we introduce Stochastic Transformer-based wORld Model (STORM), an efficient world model architecture that combines the strong sequence modeling and generation capabilities of Transformers with the stochastic nature of variational autoencoders. STORM achieves a mean human performance of $126.7\%$ on the Atari 100k benchmark, setting a new record among state-of-the-art methods that do not employ lookahead search techniques. Moreover, training an agent with $1.85$ hours of real-time interaction experience on a single NVIDIA GeForce RTX 3090 graphics card requires only $4.3$ hours, showcasing improved efficiency compared to previous methodologies.
We release our code at https://github.com/weipu-zhang/STORM.

## 1 Introduction

Deep reinforcement learning (DRL) has exhibited remarkable success across diverse domains. However, its widespread application in real-world environments is hindered by the substantial number of interactions with the environment required for achieving such success. This limitation becomes particularly challenging when dealing with broader real-world settings in e.g., unmanned and manufacturing systems [1, 2] that lack adjustable speed simulation tools. Consequently, improving the sample efficiency has emerged as a key challenge for DRL algorithms.

Popular DRL methods, including Rainbow [3] and PPO [4], suffer from low sample efficiency due to two primary reasons. Firstly, the estimation of the value function proves to be a challenging task. This

---

[*]Corresponding author

involves approximating the value function using a deep neural network (DNN) and updating it with $n$-step bootstrapped temporal difference, which naturally requires numerous iterations to converge [5]. Secondly, in scenarios where rewards are sparse, many samples exhibit similarity in terms of value functions, providing limited useful information for training and generalization of the DNN [6, 7]. This further exacerbates the challenge of improving the sample efficiency of DRL algorithms.

To address these challenges, model-based DRL algorithms have emerged as a promising approach that tackles both issues simultaneously while demonstrating significant performance gains in sample-efficient settings. These algorithms start by constructing a parameterized simulation world model of the real environment through self-supervised learning. Self-supervised learning can be implemented in various ways, such as reconstructing the original input state using a decoder [8–10], predicting actions between frames [7], or employing contrastive learning to capture the internal consistency of input states [6, 7]. These approaches provide more supervision information than conventional model-free RL losses, enhancing the feature extraction capabilities of DNNs. Subsequently, the agent's policy is improved by leveraging the experiences generated using the world model, eliminating sampling constraints and enabling faster updates to the value function compared to model-free algorithms.

However, the process of imagining with a world model involves an autoregressive process that can accumulate prediction errors over time. In situations where discrepancies arise between the imagined trajectory and the real trajectory, the agent may inadvertently pursue virtual goals, resulting in subpar performance in the real environment. To mitigate this issue, introducing random noise into the world model has been proven beneficial [9–11, 14]. Variational autoencoders, capable of automatically learning low-dimensional latent representations of high-dimensional data while incorporating reasonable random noise into the latent space, offer an ideal choice for image encoding.

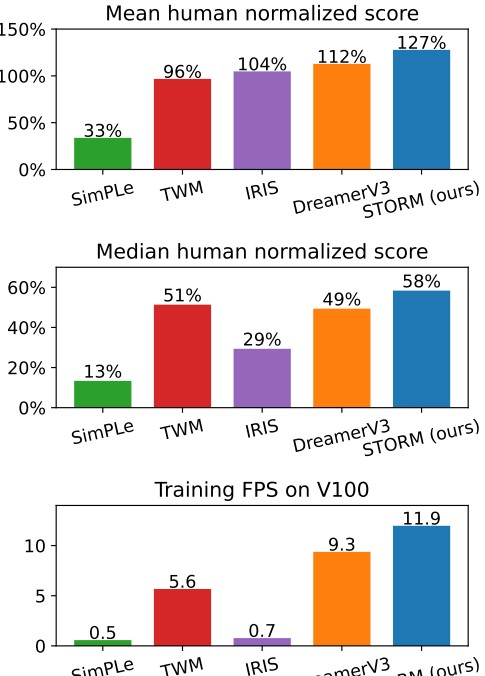

Numerous endeavors have been undertaken to construct an efficient world model. For instance, SimPLe [11] leverages LSTM [15], while DreamerV3 [10] employs GRU [16] as the sequence model. LSTM and GRU, both variants of recurrent neural networks (RNNs), excel at sequence modeling tasks. However, the recurrent nature of RNNs impedes parallelized computing, resulting in slower training speeds [17]. In contrast, the Transformer architecture [17] has lately demonstrated superior performance over RNNs in various sequence modeling and generation tasks. It overcomes the challenge of forgetting long-term dependencies and is designed for efficient parallel computing. While several attempts have been made to incorporate Transformers into the world model [12, 13, 18], these works do not fully harness the capabilities of this architecture. Furthermore, these approaches require even longer training times and fail to surpass the performance of the GRU-based DreamerV3.

Figure 1: Comparison of methods on Atari 100k. SimPLe [11] and DreamerV3 [10] employ RNNs as their world models, whereas TWM [12], IRIS [13], and STORM use Transformers. The training frames per second (FPS) results on a single NVIDIA V100 GPU are extrapolated from other graphics cards for SimPLe, TWM, and IRIS, while DreamerV3 and STORM are directly evaluated.

In this paper, we introduce the Stochastic Transformer-based wORld Model (STORM), a highly effective and efficient structure for model-based RL. STORM employs a categorical variational autoencoder (VAE) as the image encoder, enhancing the agent robustness and reducing accumulated autoregressive prediction errors. Subsequently, we incorporate the Transformer as the sequence

model, improving modeling and generation quality while accelerating training. STORM achieves a remarkable mean human normalized score of 126.7% on the challenging Atari 100k benchmark, establishing a new record for methods without resorting to lookahead search. Furthermore, training an agent with 1.85 hours of real-time interaction experience on a single NVIDIA GeForce RTX 3090 graphics card requires only 4.3 hours, demonstrating superior efficiency compared to previous methodologies. The comparison of our approach with the state-of-the-art methods is depicted in Figure 1.

## 2 Related work

Model-based DRL algorithms aim to construct a simulation model of the environment and utilize simulated experiences to improve the policy. While traditional model-based RL techniques like Dyna-Q have shown success in tabular cases [5], modeling complex environments such as video games and visual control tasks presents significant challenges. Recent advances in computing and DNNs have enabled model-based methods to learn the dynamics of the environments and start to outperform model-free methods on these tasks.

The foundation of VAE-LSTM-based world models was introduced by Ha and Schmidhuber [14] for image-based environments, demonstrating the feasibility of learning a good policy solely from generated data. SimPLe [11] applied this methodology to Atari games, resulting in substantial sample efficiency improvements compared to Rainbow [3], albeit with relatively lower performance under limited samples. The Dreamer series [8–10] also adopt this framework and showcase notable capabilities in Atari games, DeepMind Control, Minecraft, and other domains, using GRU [16] as the core sequential model. However, as discussed earlier, RNN structures suffer from slow training [17].

Recent approaches such as IRIS [13], TWM [12], and TransDreamer [18] incorporate the Transformer architecture into their world models. IRIS [13] employs VQ-VAE [19] as the encoder to map images into $4 \times 4$ latent tokens and uses a spatial-temporal Transformer [20] to capture information within and across images. However, the attention operations on a large number of tokens in the spatial-temporal structure can result in a significant training slowdown. TWM [12] adopts Transformer-XL [21] as its core architecture and organizes the sequence model in a structure similar to Decision Transformer [22], treating the observation, action, and reward as equivalent input tokens for the Transformer. Performing self-attention across different types of data may have a negative impact on the performance, and the increased number of tokens considerably slows down training. TransDreamer [18] directly replaces the GRU structure of Dreamer with Transformer. However, there is a lack of evidence demonstrating their performance in widely accepted environments or under limited sample conditions.

Other model-based RL methods such as MuZero [23], EfficientZero [7], and SpeedyZero [24] incorporate Monte Carlo tree search (MCTS) to enhance policy and achieve promising performance on the Atari 100k benchmark. Lookahead search techniques like MCTS can be employed to enhance other model-based RL algorithms, but they come with high computational demands. Additionally, certain model-free methods [6, 25–27] incorporate a self-supervised loss as an auxiliary term alongside the standard RL loss, demonstrating their effectiveness in sample-efficient settings. Additionally, recent studies [28, 29] delve deeper into model-free RL, demonstrating strong performance and high data efficiency, rivaling that of model-based methods on several benchmarks. However, since the primary objective of this paper is to enhance the world model, we do not delve further into these methods.

We highlight the distinctions between STORM and recent approaches in the world model as follows:

Table 1: Comparison between STORM and recent approaches. "Tokens" refers to the input tokens introduced to the sequence model during a single timestep. "Historical information" indicates whether the VAE reconstruction process incorporates historical data, such as the hidden states of an RNN.

| Attributes | SimPLe [11] | TWM [12] | IRIS [13] | DreamerV3 [10] | STORM (ours) |
|---|---|---|---|---|---|
| Sequence model | LSTM [15] | Transformer-XL [21] | Transformer [17] | GRU [16] | Transformer |
| Tokens | Latent | Latent, action, reward | Latent($4 \times 4$) | Latent | Latent |
| Latent representation | Binary-VAE | Categorical-VAE | VQ-VAE | Categorical-VAE | Categorical-VAE |
| Historical information | Yes | No | Yes | Yes | No |
| Agent state | Reconstructed image | Latent | Reconstructed image | Latent, hidden | Latent, hidden |
| Agent training | PPO [4] | As DreamerV2 [9] | As DreamerV2 [9] | DreamerV3 | As DreamerV3 |

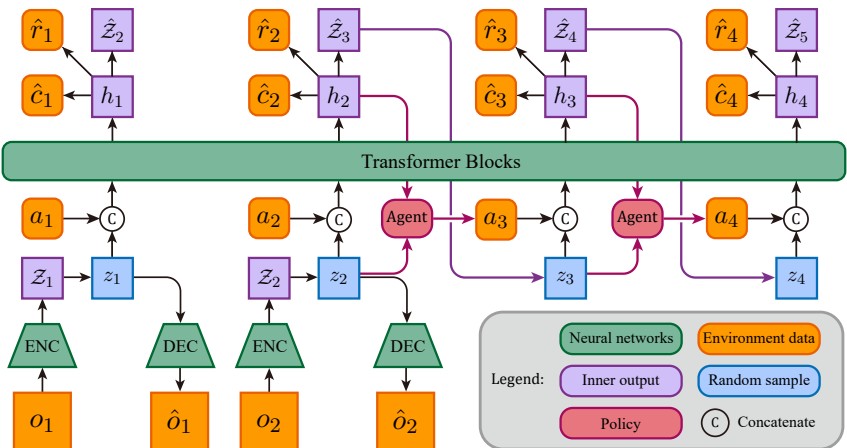

Figure 2: Structure and imagination process of STORM. The symbols used in the figure are explained in Sections 3.1 and 3.2. The Transformer blocks depict the sequence model $f_\phi$ in Equation (2). The Agent block, represented by a neural network, corresponds to $\pi_\theta(a_t|s_t)$ in Equation (6).

- SimPLe [11] and Dreamer [10] rely on RNN-based models, whereas STORM employs a GPT-like Transformer [30] as the sequence model.

- In contrast to IRIS [13] that employs multiple tokens, STORM utilizes a single stochastic latent variable to represent an image.

- STORM follows a vanilla Transformer [17] structure, while TWM [12] adopts a Transformer-XL [21] structure.

- In the sequence model of STORM, an observation and an action are fused into a single token, whereas TWM [12] treats observation, action, and reward as three separate tokens of equal importance.

- Unlike Dreamer [10] and TransDreamer [18], which incorporate hidden states, STORM reconstructs the original image without utilizing this information.

## 3  Method

### 3.1  World model learning

Our approach adheres to the established framework of model-based RL algorithms, which focus on enhancing the agent's policy by imagination [5, 9–11, 13]. We iterate through the following steps until reaching the prescribed number of real environment interactions.

S1) Gather real environment data by executing the current policy for several steps and append them to the replay buffer.

S2) Update the world model using trajectories sampled from the replay buffer.

S3) Improve the policy using imagined experiences generated by the world model, with the starting points for the imagination process sampled from the replay buffer.

At each time $t$, a data point comprises an observation $o_t$, an action $a_t$, a reward $r_t$, and a continuation flag $c_t$ (a Boolean variable indicating whether the current episode is ongoing). The replay buffer maintains a first-in-first-out queue structure, enabling the sampling of consecutive trajectories from the buffer.

Section 3.1 provides a detailed description of the architecture and training losses employed by STORM. On the other hand, Section 3.2 elaborates on the imagination process and the training methodology employed by the agent. It provides an thorough explanation of how the agent leverages the world model to simulate experiences and improve its policy.

**Model structure** The complete structure of our world model is illustrated in Figure 2. In our experiments, we focus on Atari games [31], which generate image observations $o_t$ of the environment. Modeling the dynamics of the environment directly on raw images is computationally expensive and prone to errors [7–11, 13, 23]. To address this, we leverage a VAE [32] formulated in Equation (1) to convert $o_t$ into latent stochastic categorical distributions $\mathcal{Z}_t$. Consistent with prior work [9, 10, 12], we set $\mathcal{Z}_t$ as a stochastic distribution comprising 32 categories, each with 32 classes. The encoder $(q_\phi)$ and decoder $(p_\phi)$ structures are implemented as convolutional neural networks (CNNs) [33]. Subsequently, we sample a latent variable $z_t$ from $\mathcal{Z}_t$ to represent the original observation $o_t$. Since sampling from a distribution lacks gradients for backward propagation, we apply the straight-through gradients trick [9, 34] to preserve them.

$$
\begin{aligned}
\text{Image encoder:} \quad & z_t \sim q_\phi(z_t|o_t) = \mathcal{Z}_t \\
\text{Image decoder:} \quad & \hat{o}_t = p_\phi(z_t).
\end{aligned}
\tag{1}
$$

Before entering the sequence model, we combine the latent sample $z_t$ and the action $a_t$ into a single token $e_t$ using multi-layer perceptrons (MLPs) and concatenation. This operation, denoted as $m_\phi$, prepares the inputs for the sequence model. The sequence model $f_\phi$ takes the sequence of $e_t$ as input and produces hidden states $h_t$. We adopt a GPT-like Transformer structure [30] for the sequence model, where the self-attention blocks are masked with a subsequent mask allowing $e_t$ to attend to the sequence $e_1, e_2, \ldots, e_t$. By utilizing MLPs $g_\phi^D$, $g_\phi^R$, and $g_\phi^C$, we rely on $h_t$ to predict the current reward $\hat{r}_t$, the continuation flag $\hat{c}_t$, and the next distribution $\hat{\mathcal{Z}}_{t+1}$. The formulation of this part of the world model is as follows

$$
\begin{aligned}
\text{Action mixer:} \quad & e_t = m_\phi(z_t, a_t) \\
\text{Sequence model:} \quad & h_{1:T} = f_\phi(e_{1:T}) \\
\text{Dynamics predictor:} \quad & \hat{\mathcal{Z}}_{t+1} = g_\phi^D(\hat{z}_{t+1}|h_t) \\
\text{Reward predictor:} \quad & \hat{r}_t = g_\phi^R(h_t) \\
\text{Continuation predictor:} \quad & \hat{c}_t = g_\phi^C(h_t).
\end{aligned}
\tag{2}
$$

**Loss functions** The world model is trained in a self-supervised manner, optimizing it end-to-end. The total loss function is calculated as in Equation (3) below, with fixed hyperparameters $\beta_1 = 0.5$ and $\beta_2 = 0.1$. In the equation, $B$ denotes the batch size, and $T$ denotes the batch length

$$
\mathcal{L}(\phi) = \frac{1}{BT} \sum_{n=1}^{B} \sum_{t=1}^{T} \Big[ \mathcal{L}_t^{\text{rec}}(\phi) + \mathcal{L}_t^{\text{rew}}(\phi) + \mathcal{L}_t^{\text{con}}(\phi) + \beta_1 \mathcal{L}_t^{\text{dyn}}(\phi) + \beta_2 \mathcal{L}_t^{\text{rep}}(\phi) \Big].
\tag{3}
$$

The individual components of the loss function are defined as follows: $\mathcal{L}_t^{\text{rec}}(\phi)$ represents the reconstruction loss of the original image, $\mathcal{L}_t^{\text{rew}}(\phi)$ represents the prediction loss of the reward, and $\mathcal{L}_t^{\text{con}}(\phi)$ represents the prediction loss of the continuation flag.

$$
\begin{aligned}
\mathcal{L}_t^{\text{rec}}(\phi) &= ||\hat{o}_t - o_t||_2 & \text{(4a)} \\
\mathcal{L}_t^{\text{rew}}(\phi) &= \mathcal{L}^{\text{sym}}(\hat{r}_t, r_t) & \text{(4b)} \\
\mathcal{L}_t^{\text{con}}(\phi) &= c_t \log \hat{c}_t + (1 - c_t) \log(1 - \hat{c}_t). & \text{(4c)}
\end{aligned}
$$

Additionally, $\mathcal{L}^{\text{sym}}$ in Equation (4b) denotes the symlog two-hot loss, as described in [10]. This loss function transforms the regression problem into a classification problem, ensuring consistent loss scaling across different environments.

The losses $\mathcal{L}_t^{\text{dyn}}(\phi)$ and $\mathcal{L}_t^{\text{rep}}(\phi)$ are expressed as Kullback–Leibler (KL) divergences but differ in their gradient backward and weighting. The dynamics loss $\mathcal{L}_t^{\text{dyn}}(\phi)$ guides the sequence model in predicting the next distribution, while the representation loss $\mathcal{L}_t^{\text{rep}}(\phi)$ allows the output of the encoder to be weakly influenced by the sequence model's prediction. This ensures that the learning of distributional dynamics is not excessively challenging.

$$
\begin{aligned}
\mathcal{L}_t^{\text{dyn}}(\phi) &= \max\big(1, \text{KL}\big[\text{sg}(q_\phi(z_{t+1}|o_{t+1})) \,||\, g_\phi^D(\hat{z}_{t+1}|h_t)\big]\big) & \text{(5a)} \\
\mathcal{L}_t^{\text{rep}}(\phi) &= \max\big(1, \text{KL}\big[q_\phi(z_{t+1}|o_{t+1}) \,||\, \text{sg}(g_\phi^D(\hat{z}_{t+1}|h_t))\big]\big) & \text{(5b)}
\end{aligned}
$$

where $\text{sg}(\cdot)$ denotes the operation of stop-gradients.

## 3.2 Agent learning

The agent's learning is solely based on the imagination process facilitated by the world model, as illustrated in Figure 2. To initiate the imagination process, a brief contextual trajectory is randomly selected from the replay buffer, and the initial posterior distribution $\mathcal{Z}_t$ is computed. During inference, rather than sampling directly from the posterior distribution $\mathcal{Z}_t$, we sample $z_t$ from the prior distribution $\hat{\mathcal{Z}}_t$. To accelerate the inference, we employ the KV cache technique [35] within the Transformer structure.

The agent's state is formed by concatenating $z_t$ and $h_t$, as shown below:

$$\text{State:} \quad s_t = [z_t, h_t]$$

$$\text{Critic:} \quad V_\psi(s_t) \approx \mathbb{E}_{\pi_\theta, p_\phi}\left[\sum_{k=0}^{\infty} \gamma^k r_{t+k}\right] \tag{6}$$

$$\text{Actor:} \quad a_t \sim \pi_\theta(a_t|s_t).$$

We adopt the actor learning settings from DreamerV3 [10]. The complete loss of the actor-critic algorithm is described by Equation (7), where $\hat{r}_t$ corresponds to the reward predicted by the world model, and $\hat{c}_t$ represents the predicted continuation flag:

$$\mathcal{L}(\theta) = \frac{1}{BL}\sum_{n=1}^{B}\sum_{t=1}^{L}\left[-\text{sg}\left(\frac{G_t^\lambda - V_\psi(s_t)}{\max(1, S)}\right)\ln \pi_\theta(a_t|s_t) - \eta H\big(\pi_\theta(a_t|s_t)\big)\right] \tag{7a}$$

$$\mathcal{L}(\psi) = \frac{1}{BL}\sum_{n=1}^{B}\sum_{t=1}^{L}\left[\Big(V_\psi(s_t) - \text{sg}\big(G_t^\lambda\big)\Big)^2 + \Big(V_\psi(s_t) - \text{sg}\big(V_{\psi^{\text{EMA}}}(s_t)\big)\Big)^2\right] \tag{7b}$$

where $H(\cdot)$ denotes the entropy of the policy distribution, while constants $\eta$ and $L$ represent the coefficient for entropy loss and the imagination horizon, respectively. The $\lambda$-return $G_t^\lambda$ [5, 10] is recursively defined as follows

$$G_t^\lambda \doteq r_t + \gamma c_t\left[(1-\lambda)V_\psi(s_{t+1}) + \lambda G_{t+1}^\lambda\right] \tag{8a}$$

$$G_L^\lambda \doteq V_\psi(s_L). \tag{8b}$$

The normalization ratio $S$ utilized in the actor loss (7a) is defined in Equation (9), which is computed as the range between the 95th and 5th percentiles of the $\lambda$-return $G_t^\lambda$ across the batch [10]

$$S = \text{percentile}(G_t^\lambda, 95) - \text{percentile}(G_t^\lambda, 5). \tag{9}$$

To regularize the value function, we maintain the exponential moving average (EMA) of $\psi$. The EMA is defined in Equation (10), where $\psi_t$ represents the current critic parameters, $\sigma$ is the decay rate, and $\psi_{t+1}^{\text{EMA}}$ denotes the updated critic parameters. This regularization technique aids in stabilizing training and preventing overfitting

$$\psi_{t+1}^{\text{EMA}} = \sigma\psi_t^{\text{EMA}} + (1-\sigma)\psi_t. \tag{10}$$

## 4 Experiments

We evaluated the performance of STORM on the widely-used benchmark for sample-efficient RL, Atari 100k [31]. For detailed information about the benchmark, evaluation methodology, and the baselines used for comparison, please refer to Section 4.1. The comprehensive results for the Atari 100k games are presented in Section 4.2.

### 4.1 Benchmark and baselines

Atari 100k consists of 26 different video games with discrete action dimensions of up to 18. The 100k sample constraint corresponds to 400k actual game frames, taking into account frame skipping (4 frames skipped) and repeated actions within those frames. This constraint corresponds to

approximately 1.85 hours of real-time gameplay. The agent's human normalized score $\tau = \frac{A-R}{H-R}$ is calculated based on the score $A$ achieved by the agent, the score $R$ obtained by a random policy, and the average score $H$ achieved by a human player in a specific environment. To determine the human player's performance $H$, a player is allowed to become familiar with the game under the same sample constraint.

To demonstrate the efficiency of our proposed world model structure, we compare it with model-based DRL algorithms that share a similar training pipeline, as discussed in Section 2. However, similarly to [10, 12, 13], we do not directly compare our results with lookahead search methods like MuZero [23] and EfficientZero [7], as our primary goal is to refine the world model itself. Nonetheless, lookahead search techniques can be combined with our method in the future to further enhance the agent's performance.

## 4.2 Results on Atari 100k

Detailed results for each environment can be found in Table 2, and the corresponding performance curve is presented in Appendix A due to space limitations. In our experiments, we trained STORM using 5 different seeds and saved checkpoints every $2,500$ sample steps. We assessed the agent's performance by conducting 20 evaluation episodes for each checkpoint and computed the average score. The result reported in Table 2 is the average of the scores attained using the final checkpoints.

STORM demonstrates superior performance compared to previous methods in environments where the key objects related to rewards are large or multiple, such as *Amidar*, *MsPacman*, *Chopper Command*, and *Gopher*. This advantage can be attributed to the attention mechanism, which explicitly preserves the history of these moving objects, allowing for an easy inference of their speed and direction information, unlike RNN-based methods. However, STORM faces challenges when handling a single small moving object, as observed in *Pong* and *Breakout*, due to the nature of autoencoders. Moreover, performing attention operations under such circumstances can potentially harm performance, as the randomness introduced by sampling may excessively influence the attention weights.

Table 2: Game scores and overall human-normalized scores on the 26 games in the Atari 100k benchmark. Following the conventions of [9], scores that are the highest or within 5% of the highest score are highlighted in bold.

| Game | Random | Human | SimPLe [11] | TWM [12] | IRIS [13] | DreamerV3 [10] | STORM (ours) |
|---|---|---|---|---|---|---|---|
| Alien | 228 | 7128 | 617 | 675 | 420 | **959** | **984** |
| Amidar | 6 | 1720 | 74 | 122 | 143 | 139 | **205** |
| Assault | 222 | 742 | 527 | 683 | **1524** | 706 | 801 |
| Asterix | 210 | 8503 | **1128** | **1116** | 854 | 932 | 1028 |
| Bank Heist | 14 | 753 | 34 | 467 | 53 | **649** | **641** |
| Battle Zone | 2360 | 37188 | 4031 | 5068 | **13074** | 12250 | **13540** |
| Boxing | 0 | 12 | 8 | **78** | 70 | **78** | **80** |
| Breakout | 2 | 30 | 16 | 20 | **84** | 31 | 16 |
| Chopper Command | 811 | 7388 | 979 | 1697 | 1565 | 420 | **1888** |
| Crazy Climber | 10780 | 35829 | 62584 | 71820 | 59234 | **97190** | 66776 |
| Demon Attack | 152 | 1971 | 208 | 350 | **2034** | 303 | 165 |
| Freeway | 0 | 30 | 17 | 24 | **31** | 0 | **34** |
| Freeway w/o traj | 0 | 30 | 17 | 24 | **31** | 0 | 0 |
| Frostbite | 65 | 4335 | 237 | **1476** | 259 | 909 | 1316 |
| Gopher | 258 | 2413 | 597 | 1675 | 2236 | 3730 | **8240** |
| Hero | 1027 | 30826 | 2657 | 7254 | 7037 | **11161** | **11044** |
| James Bond | 29 | 303 | 101 | 362 | 463 | 445 | **509** |
| Kangaroo | 52 | 3035 | 51 | 1240 | 838 | **4098** | **4208** |
| Krull | 1598 | 2666 | 2204 | 6349 | 6616 | 7782 | **8413** |
| Kung Fu Master | 256 | 22736 | 14862 | 24555 | 21760 | 21420 | **26182** |
| Ms Pacman | 307 | 6952 | 1480 | 1588 | 999 | 1327 | **2673** |
| Pong | -21 | 15 | 13 | **19** | 15 | **18** | 11 |
| Private Eye | 25 | 69571 | 35 | 87 | 100 | 882 | **7781** |
| Qbert | 164 | 13455 | 1289 | 3331 | 746 | 3405 | **4522** |
| Road Runner | 12 | 7845 | 5641 | 9109 | 9615 | 15565 | **17564** |
| Seaquest | 68 | 42055 | 683 | **774** | 661 | 618 | 525 |
| Up N Down | 533 | 11693 | 3350 | **15982** | 3546 | 7667 | 7985 |
| Human Mean | 0% | 100% | 33% | 96% | 105% | 112% | **126.7%** |
| Human Median | 0% | 100% | 13% | 51% | 29% | 49% | **58.4%** |

# 5 Ablation studies

In our experiments, we have observed that the design and configuration choices of the world model and the agent can have significant impacts on the final results. To further investigate this, we conduct ablation studies on the design and configuration of the world model in Section 5.1, as well as on the agent's design in Section 5.2. Additionally, we propose a novel approach to enhancing the exploration efficiency through the imagination capability of the world model using a single demonstration trajectory, which is explained in Section 5.3.

## 5.1 World model design and configuration

The RNN-based world models utilized in SimPLe [11] and Dreamer [9, 10] can be formulated clearly using variational inference over time. However, the non-recursive Transformer-based world model does not align with this practice and requires manual design. Figure 3a shows alternative structures and their respective outcomes. In the "Decoder at rear" configuration, we employ $z_t \sim \hat{\mathcal{Z}}_t$ instead of $z_t \sim \mathcal{Z}_t$ for reconstructing the original observation and calculating the loss. The results indicate that the reconstruction loss should be applied directly to the output of the encoder rather than relying on the sequence model. In the "Predictor at front" setup, we utilize $z_t$ as input for $g_\phi^R(\cdot)$ and $g_\phi^C(\cdot)$ in Equation (2), instead of $h_t$. These findings indicate that, while this operation has minimal impact on the final performance for tasks where the reward can be accurately predicted from a single frame (in e.g., *Pong*), it leads to a performance drop on tasks that require several contextual frames to predict the reward accurately (in e.g., *Ms. Pacman*).

By default, we configure our Transformer with 2 layers, which is significantly smaller than the 10 layers used in IRIS [13] and TWM [12]. Figure 3b presents the varied outcomes obtained by increasing the number of Transformer layers. The results reveal that increasing the layer count does not have a positive impact on the final performance. However, in the case of the game *Pong*, even when the sample limit is increased from 100k to 400k, the agent still achieves the maximum reward in this environment regardless of whether a 4-layer or 6-layer Transformer is employed. This scaling discrepancy, which differs from the success observed in other fields [36–38], may be attributed to three reasons. Firstly, due to the minor difference between adjacent frames and the presence of residual connections in the Transformer structure [17], predicting the next frame may not require a complex model. Secondly, training a large model naturally requires a substantial amount of data, yet the Atari 100k games neither provide images from diverse domains nor offer sufficient samples for training a larger model. Thirdly, the world model is trained end-to-end, and the representation loss $\mathcal{L}^{\text{rep}}$ in Equation (5b) directly influences the image encoder. The encoder may be overly influenced when tracking the output of a large sequence model.

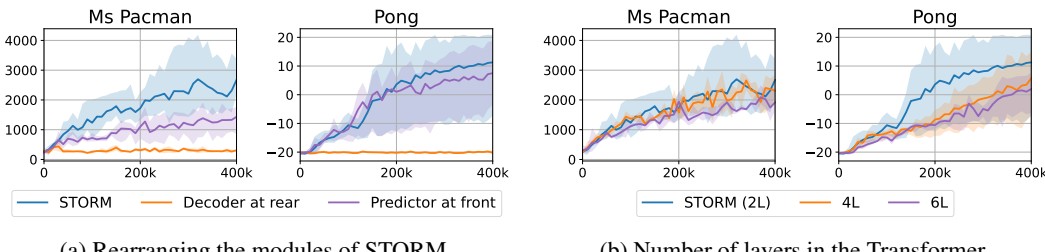

(a) Rearranging the modules of STORM.          (b) Number of layers in the Transformer.

Figure 3: Ablation studies on the design and configuration of the STORM's world model.

## 5.2 Selection of the agent's state

The choice of the agent's state $s_t$ offers several viable options: $\hat{o}_t$ [13], $h_t$, $z_t$ (as in TWM [12]), or the combination $[h_t, z_t]$ (as demonstrated by Dreamer, [9, 10]). In the case of STORM, we employ $s_t = [h_t, z_t]$, as in Equation (6). Ablation studies investigating the selection of the agent's state are presented in Figure 4. The results indicate that, in environments where a good policy requires contextual information, such as in *Ms. Pacman*, the inclusion of $h_t$ leads to improved performance. However, in other environments like *Pong* and *Kung Fu Master*, this inclusion does not yield a significant difference. When solely utilizing $h_t$ in environments that evolve with the agent's policy,

like *Pong*, the agent may exhibit behaviors similar to catastrophic forgetting [39] due to the non-stationary and inaccurate nature of the world model. Consequently, the introduction of randomness through certain distributions like $\mathcal{Z}_t$ proves to be beneficial.

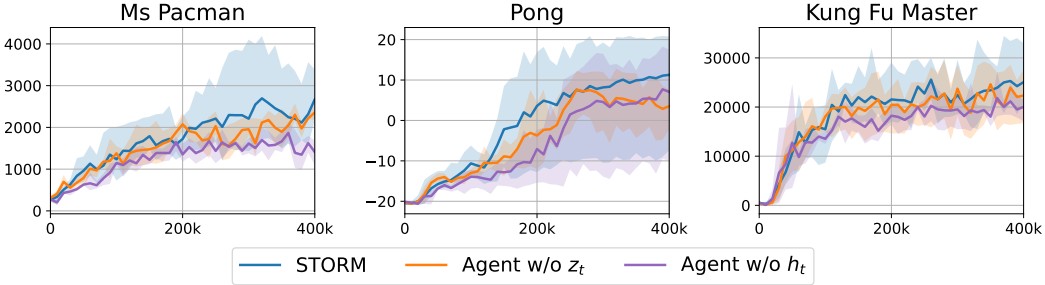

Figure 4: Ablation studies on the selection of the agent's state.

## 5.3 Impact of the demonstration trajectory

The inclusion of a demonstration trajectory is a straightforward implementation step when using a world model, and it is often feasible in real-world settings. Figure 5 showcases the impact of incorporating a single demonstration trajectory in the replay buffer. Details about the provided trajectory can be found in Appendix D. In environments with sparse rewards, adding a trajectory can improve the robustness, as observed in *Pong*, or the performance, as seen in *Freeway*. However, in environments with dense rewards like *Ms Pacman*, including a trajectory may hinder the policy improvement of the agent.

*Freeway* serves as a prototypical environment characterized by challenging exploration but a simple policy. To receive a reward, the agent must take the "up" action approximately 70 times in a row, but it quickly improves its policy once the first reward is obtained. Achieving the first reward is extremely challenging if the policy is initially set as a uniform distribution of actions. In the case of TWM [12], the entropy normalization technique is employed across all environments, while IRIS [13] specifically reduces the temperature of the Boltzmann exploration strategy for *Freeway*. These tricks are critical in obtaining the first reward in this environment. It is worth noting that even for most humans, playing exploration-intensive games without prior knowledge is challenging. Typically, human players require instructions about the game's objectives or watch demonstrations from teaching-level or expert players to gain initial exploration directions. Inspired by this observation, we aim to directly incorporate a demonstration trajectory to train the world model and establish starting points for imagination. By leveraging a sufficiently robust world model, the utilization of limited offline information holds the potential to surpass specially designed curiosity-driven exploration strategies in the future.

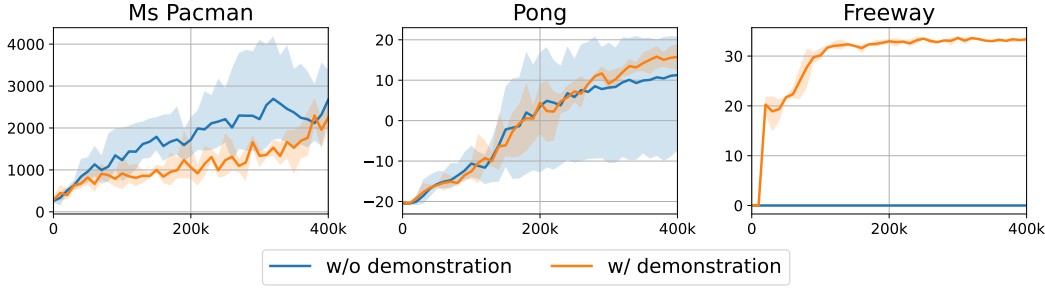

Figure 5: Ablations studies on adding a demonstration trajectory to the replay buffer.

As an integral part of our methodology, we integrate a single trajectory from *Freeway* into our extensive results. Furthermore, to ensure fair comparisons in future research, we provide the results without incorporating *Freeway*'s trajectory in Table 2 and Figure 6. It is important to highlight that even without this trajectory, our approach consistently outperforms previous methods, attaining a mean human normalized score of $122.3\%$.

# 6 Conclusions and limitations

In this work, we introduce STORM, an efficient world model architecture for model-based RL, surpassing previous methods in terms of both performance and training efficiency. STORM harnesses the powerful sequence modeling and generation capabilities of the Transformer structure while fully exploiting its parallelizable training advantages. The improved efficiency of STORM broadens its applicability across a wider range of tasks while reducing computational costs.

Nevertheless, it is important to acknowledge certain limitations. Firstly, both the world model of STORM and the compared baselines are trained in an end-to-end fashion, where the image encoder and sequence model undergo joint optimization. As a result, the world model must predict its own internal output, introducing additional non-stationarity into the optimization process and potentially impeding the scalability of the world model. Secondly, the starting points for imagination are uniformly sampled from the replay buffer, while the agent is optimized using an on-policy actor-critic algorithm. Although acting in the world model is performed on-policy, the corresponding on-policy distribution $\mu(s)$ for these starting points is not explicitly considered, despite its significance in the policy gradient formulation: $\nabla J(\theta) \propto \sum_s \mu(s) \sum_a q_{\pi_\theta}(s,a) \nabla \pi_\theta(a|s)$ [5].

## Acknowledgments and Disclosure of Funding

We would like to thank anonymous reviewers for their constructive comments. The work was supported partially by the National Key R&D Program of China under Grant 2021YFB1714800, partially by the National Natural Science Foundation of China under Grants 62173034, U23B2059, 61925303, 62088101, and partially by the Chongqing Natural Science Foundation under Grant 2021ZX4100027.

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

# A  Atari 100k curves

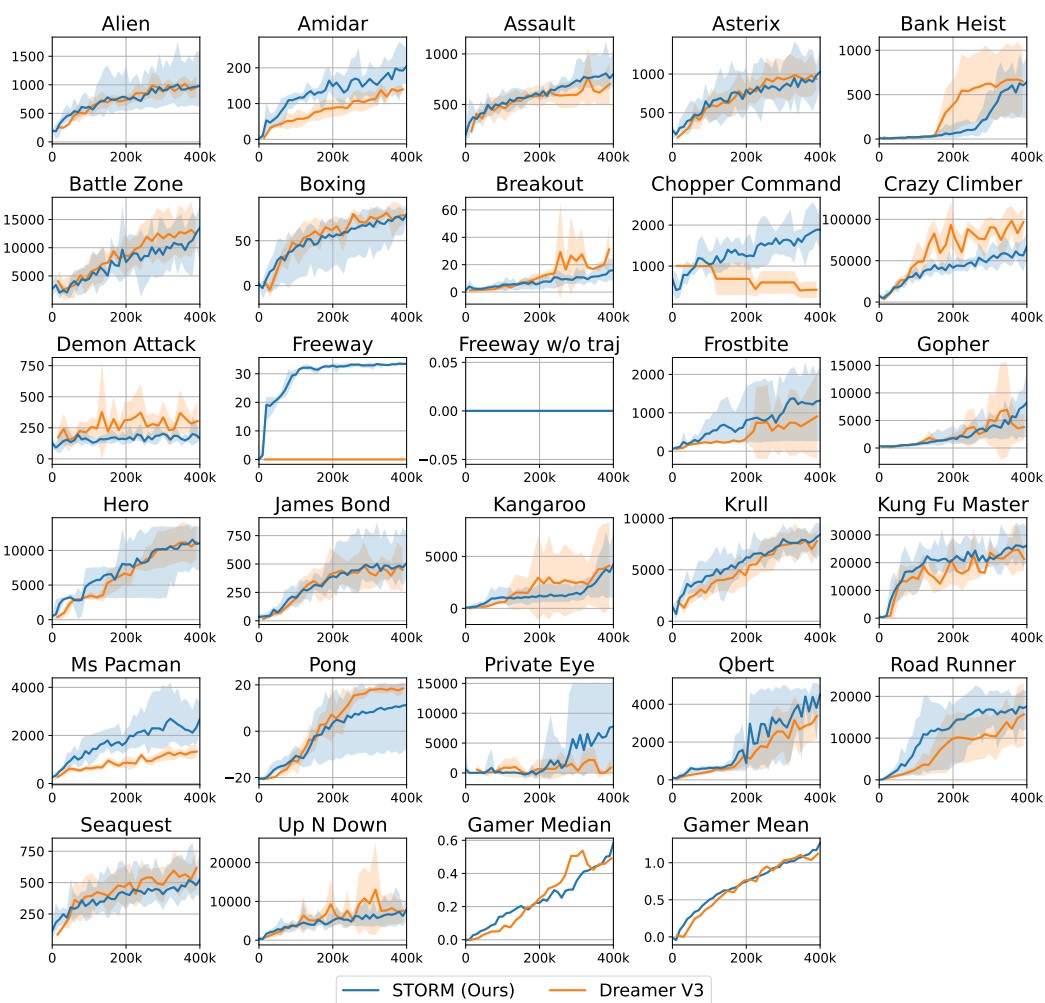

Figure 6: Performance comparison on the Atari 100k benchmark. Our method is represented in blue, while DreamerV3 [10] is in orange. The solid line represents the average result over 5 seeds, and the filled area indicates the range between the maximum and minimum results across these 5 seeds.

# B   Details of model structure

Table 3: Structure of the image encoder. The size of the submodules is omitted and can be derived from the shape of the tensors. ReLU refers to the rectified linear units used for activation, while Linear represents a fully-connected layer. Flatten and Reshape operations are employed to alter the indexing method of the tensor while preserving the data and their original order. Conv denotes a CNN layer [33], characterized by kernel $= 4$, stride $= 2$, and padding $= 1$. BN denotes the batch normalization layer[40].

| Submodule | Output tensor shape |
|---|---|
| Input image ($o_t$) | $3 \times 64 \times 64$ |
| Conv1 + BN1 + ReLU | $32 \times 32 \times 32$ |
| Conv2 + BN2 + ReLU | $64 \times 16 \times 16$ |
| Conv3 + BN3 + ReLU | $128 \times 8 \times 8$ |
| Conv4 + BN4 + ReLU | $256 \times 4 \times 4$ |
| Flatten | 4096 |
| Linear | 1024 |
| Reshape (produce $\mathcal{Z}_t$) | $32 \times 32$ |

Table 4: Structure of the image decoder. DeConv denotes a transpose CNN layer [41], characterized by kernel $= 4$, stride $= 2$, and padding $= 1$.

| Submodule | Output tensor shape |
|---|---|
| Random sample ($z_t$) | $32 \times 32$ |
| Flatten | 1024 |
| Linear + BN0 + ReLU | 4096 |
| Reshape | $256 \times 4 \times 4$ |
| DeConv1 + BN1 + ReLU | $128 \times 8 \times 8$ |
| DeConv2 + BN2 + ReLU | $64 \times 16 \times 16$ |
| DeConv3 + BN3 + ReLU | $32 \times 32 \times 32$ |
| DeConv4 (produce $\hat{o}_t$) | $3 \times 64 \times 64$ |

Table 5: Action mixer $e_t = m_\phi(z_t, a_t)$. Concatenate denotes combining the last dimension of two tensors and merging them into one new tensor. The variable $A$ represents the action dimension, which ranges from 3 to 18 across different games. $D$ denotes the feature dimension of the Transformer. LN is an abbreviation for layer normalization [42].

| Submodule | Output tensor shape |
|---|---|
| Random sample ($z_t$), Action ($a_t$) | $32 \times 32, A$ |
| Reshape and concatenate | $1024 + A$ |
| Linear1 + LN1 + ReLU | $D$ |
| Linear2 + LN2 (output $e_t$) | $D$ |

Table 6: Positional encoding module. $w_{1:T}$ is a learnable parameter matrix with shape $T \times D$, and $T$ refers to the sequence length.

| Submodule | Output tensor shape |
|---|---|
| Input ($e_{1:T}$) | |
| Add ($e_{1:T} + w_{1:T}$) | $T \times D$ |
| LN | |

Table 7: Transformer block. Dropout mechanism [43] can prevent overfitting.

| Submodule | Module alias | Output tensor shape |
|---|---|---|
| Input features (label as $x_1$) | | $T \times D$ |
| Multi-head self attention | | |
| Linear1 + Dropout($p$) | MHSA | $T \times D$ |
| Residual (add $x_1$) | | |
| LN1 (label as $x_2$) | | |
| Linear2 + ReLU | | $T \times 2D$ |
| Linear3 + Dropout($p$) | FFN | $T \times D$ |
| Residual (add $x_2$) | | $T \times D$ |
| LN2 | | $T \times D$ |

Table 8: Transformer based sequence model $h_{1:T} = f_\phi(e_{1:T})$. Positional encoding is explained in Table 6 and Transformer block is explained in Table 7.

| Submodule | Output tensor shape |
|---|---|
| Input ($e_{1:T}$) | |
| Positional encoding | |
| Transformer blocks $\times K$ | $T \times D$ |
| Output ($h_{1:T}$) | |

Table 9: Pure MLP structures. A 1-layer MLP corresponds to a fully-connected layer. 255 is the size of the bucket of symlog two-hot loss [10].

| Module name | Symbol | MLP layers | Input/ MLP hidden/ Output dimension |
|---|---|---|---|
| Dynamics head | $g_\phi^D$ | 1 | D/ -/ 1024 |
| Reward predictor | $g_\phi^R$ | 3 | D/ D/ 255 |
| Continuation predictor | $g_\phi^C$ | 3 | D/ D/ 1 |
| Policy network | $\pi_\theta(a_t|s_t)$ | 3 | D/ D/ A |
| Critic network | $V_\psi(s_t)$ | 3 | D/ D/ 255 |

# C Hyperparameters

Table 10: Hyerparameters. Note that the environment will provide a "done" signal when losing a life, but will continue running until the actual reset occurs. This life information configuration aligns with the setup used in IRIS [13]. Regarding data sampling, each time we sample $B_1$ trajectories of length $T$ for world model training, and sample $B_2$ trajectories of length $C$ for starting the imagination process.

| Hyperparameter | Symbol | Value |
|---|---|---|
| Transformer layers | $K$ | 2 |
| Transformer feature dimension | $D$ | 512 |
| Transformer heads | - | 8 |
| Dropout probability | $p$ | 0.1 |
| World model training batch size | $B_1$ | 16 |
| World model training batch length | $T$ | 64 |
| Imagination batch size | $B_2$ | 1024 |
| Imagination context length | $C$ | 8 |
| Imagination horizon | $L$ | 16 |
| Update world model every env step | - | 1 |
| Update agent every env step | - | 1 |
| Environment context length | - | 16 |
| Gamma | $\gamma$ | 0.985 |
| Lambda | $\lambda$ | 0.95 |
| Entropy coefficiency | $\eta$ | $3 \times 10^{-4}$ |
| Critic EMA decay | $\sigma$ | 0.98 |
| Optimizer | - | Adam [44] |
| World model learning rate | - | $1.0 \times 10^{-4}$ |
| World model gradient clipping | - | 1000 |
| Actor-critic learning rate | - | $3.0 \times 10^{-5}$ |
| Actor-critic gradient clipping | - | 100 |
| Gray scale input | - | False |
| Frame stacking | - | False |
| Frame skipping | - | 4 (max over last 2 frames) |
| Use of life information | - | True |

# D   Demonstration trajectory information

Table 11: To account for frame skipping, the frame count is multiplied by 4. These trajectories were gathered using pre-trained DQN agents [45].

| Game | Episode return | Frames |
|------|----------------|--------|
| Ms Pacman | 5860 | $1612 \times 4$ |
| Pong | 18 | $2079 \times 4$ |
| Freeway | 27 | $2048 \times 4$ |

# E Computational cost details and comparison

Table 12: Computational comparison. In the V100 column, an item marked with a star indicates extrapolation based on other graphics cards, while items without a star are tested using actual devices. The extrapolation method employed aligns with the setup used in DreamerV3 [10], where it assumes the P100 is twice as slow and the A100 is twice as fast.

| Method | Original computing resource | V100 hours |
|---|---|---|
| SimPLe [11] | NVIDIA P100, 20 days | 240* |
| TWM [12] | NVIDIA A100, 10 hours
NVIDIA GeForce RTX 3090, 12.5 hours | 20* |
| IRIS [13] | NVIDIA A100, 7 days for two runs | 168* |
| DreamerV3 [10] | NVIDIA V100, 12 hours | 12 |
| STORM | NVIDIA GeForce RTX 3090, 4.3 hours | 9.3 |

# F    Atari video predictions

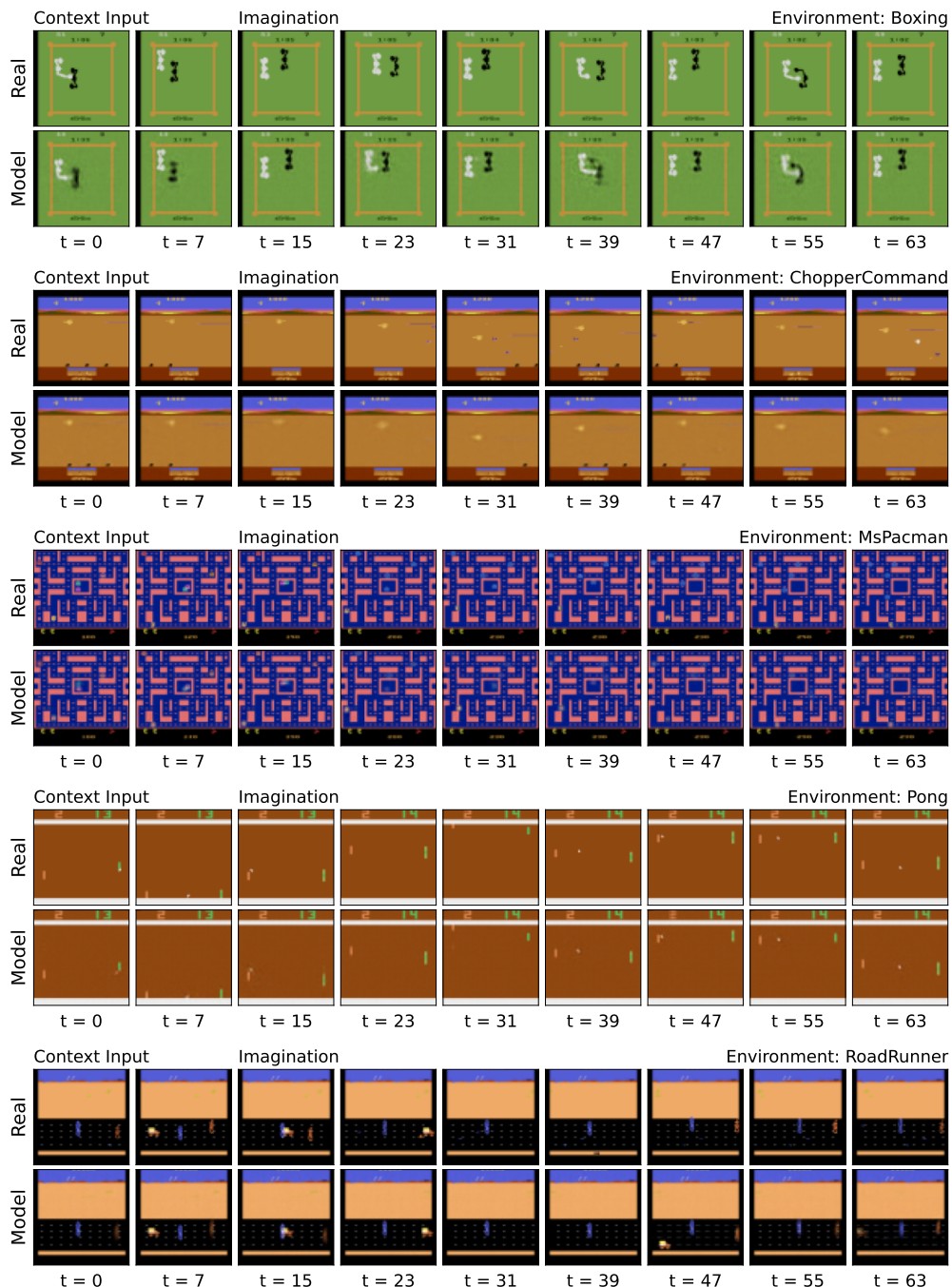

Figure 7: Multi-step predictions on several environments in Atari games. The world model utilizes 8 observations and actions as contextual input, enabling the imagination of future events spanning 56 frames in an auto-regressive manner.

