# OpenReview forum: "STORM: Efficient Stochastic Transformer based World Models for Reinforcement Learning"
_NeurIPS.cc/2023/Conference — NeurIPS 2023 poster_

### Official Review · Reviewer_E5z3 · 2023-06-27

**Soundness:** 3 good
**Presentation:** 3 good
**Contribution:** 3 good
**Rating:** 5
**Confidence:** 4

**Summary:**

The paper introduces the Stochastic Transformer-based wORld Model (STORM), an efficient world model architecture. STORM proposes to encode image inputs using a stochastic variational autoencoder, and predicts latent state using a GPT-like sequential model. It then trains dynamics and policy based on the outputs of these stochastic variational autoencoder and sequential model. The authors conducted experimental comparisons with some classic baseline methods in the Atari100k environment. The results indicate that STORM has a faster speed and achieves better performance.

**Strengths:**

1. Although the individual components proposed by STORM have been introduced before, the authors designed a solid structure to combine these parts into an effective method.
2. The experimental results demonstrate the effectiveness of STORM, and more importantly, they improve the time efficiency of Model-Based Reinforcement Learning methods.

**Weaknesses:**

1. In the experimental section, please add a comparison with Speedy Zero [1]. Speedy Zero is also a model-based RL method  which is proposed recently and has also achieved good time efficiency and performance on Atari.
2. Beyond Atari, can STORM be applied to other tasks, such as MuJoCo, DMC, MetaWorld, etc.? Dreamerv3 can achieve very good results in a wide range of different environments. If STORM cannot, the practical significance of this paper will be greatly reduced.
3. It would be better if further analysis could be provided on the benefits of using a stochastic variational autoencoder.

Reference:

[1]. Mei et.al. "SpeedyZero: Mastering Atari with Limited Data and Time." In ICLR 2023 https://openreview.net/forum?id=Mg5CLXZgvLJ.

**Questions:**

See weakness above

**Limitations:**

The generalizability of STORM could be further studied.

---

> ### Author Rebuttal · Authors · 2023-08-06
>
> Thanks for your effort spent reviewing our paper and providing many valuable suggestions. We will include the suggestions and the pointed concurrent work in the revised paper. Below, we want to address the main concerns raised in the review.
>
> - Response to **(1)**: We appreciate your observation regarding the missing paper, which we will include in the related work section **(Line 117)** rather than the experiments part for the reasons we elaborate next. As stated in **Section 2 (Lines 117-123)** and **Section 4.1 (Lines 221-226)**, the design and experimental validation of STORM have mainly followed the approach of previous works (DreamerV3, IRIS, TWM). Similarly to these works, we do not directly compare our results with **lookahead search based methods** such as MuZero, EfficientZero, and SpeedyZero, as our primary goal is to refine the world model itself. Nonetheless, lookahead search techniques can be combined with our method in the future to further enhance the agent's performance.
> - Response to **(2)**: Thanks for your suggestion on carrying out additional experiments using other benchmarks, which is interesting and indeed can further strengthen the contribution of the paper. Nonetheless, as explained in our **Response 1** in the **Author Rebuttal** above, the additional computing resources and time required for training and validation of STORM do not seem a feasible plan at present. We will explore it in the future. Please kindly refer to our **Response 1** in the **Author Rebuttal** for more details.
> - Response to **(3)**:  The use of stochastic representation is prevalent in reconstruction-based model-based RL algorithms, including STORM, Dreamer, TWM, and SimPLe. In our early experiments, we found that employing deterministic features to represent the observations (using e.g., a vanilla autoencoder) and replacing the KLDiv dynamics loss with L2 loss results in a significant inconsistency between the reconstructed environment and the original environment after a few steps. In contrast, a stochastic representation maintains stable reconstruction over a larger extended period (**Figure 7** in **Appendix F**). This phenomenon has also been observed by SimPLe [1] (in their Appendix A, ablations on models). Certainly, it should be remarked that this approach has limitations, for instance, contrastive learning methods, which have been proven effective for RL in EfficientZero/CURL, cannot be directly applied to STORM.
>
> ---
>
> **References**
>
> [1] Kaiser, Lukasz, et al. "Model-based reinforcement learning for atari." *arXiv preprint arXiv:1903.00374* (2019).

---

> > ### Comment · Reviewer_E5z3 · 2023-08-20
> >
> > Thank you for the author's response. Due to the lack of experiments in more environments, it's hard for me to judge the generalization ability of STORM. Therefore, I will maintain my score.

---

> > > ### Author Response · Authors · 2023-08-20
> > >
> > > We extend our appreciation for your careful consideration of and feedback on our rebuttal. However, we firmly posit that the Atari100k benchmark is apt for assessing STORM's generalization capacity.
> > >
> > > 1) It's noteworthy that a multitude of antecedent works, including IRIS, TWM, SimPLe (our baselines), EfficientZero, and SpeedyZero, have all elected to conduct their empirical investigations leveraging the Atari100k benchmark only. We posit that this historical precedent not only bolsters the credibility of the benchmark but also underscores its relevance to our present study. We believe that the utilization of this benchmark should not be deemed a glaring limitation of our current endeavor.
> > >
> > > 2) The support bestowed upon DreamerV3 by DeepMind affords them a resource advantage that surpasses the norm for most researchers. In contrast, our present circumstances constrain our capacity to conduct a broader spectrum of experiments—a fact that we acknowledge with a measure of regret.
> > >
> > > As highlighted in our `Response to Weakness 2` addressed to `Reviewer rfCq`, the phenomenon of two algorithms achieving identical global human mean/median scores while excelling in distinct environments is a common occurrence. STORM, in this context, emerges as a potential catalyst. Its demonstrated ability to excel in specific environments, as detailed in Section 4.2, adds a distinctive reference option for practitioners. This novel dimension, we believe, can enrich practical applications.
> > >
> > > We extend our gratitude for dedicating your time to our rebuttal again. We will appreciate it if you could reconsider assessing the paper.

---

> ### Comment · Area_Chair_8ViK · 2023-08-17
>
> Dear Reviewer,
>
> The author has posted their rebuttal, but you have not yet posted your response. Please post your thoughts after reading the rebuttal and other reviews as soon as possible. All reviewers are requested to post this after-rebuttal-response.

---

### Official Review · Reviewer_rfCq · 2023-07-05

**Soundness:** 3 good
**Presentation:** 3 good
**Contribution:** 2 fair
**Rating:** 7
**Confidence:** 5

**Summary:**

This paper proposes a world model architecture (STORM) to train RL agents in imagination. The world model is composed of an autoencoder with categorical latents and a Transformer. These modules are trained jointly with a reconstruction loss, a next latent state prediction loss, as well as reward, episode termination, and representation losses. Experiments in the Atari 100k benchmark indicate that the approach is effective.

**Strengths:**

- The method is technically sound with empirical results to back its effectiveness (outperforms other methods based on learning in imagination).
- It is faster to train than similar methods (e.g. twice as fast as TWM).
- The paper is well-written and easy to follow.

**Weaknesses:**

- The method lacks novelty and the incremental improvements over previous work are not properly explored.
    - STORM is a variant of TWM with two minor modifications: (1) a vanilla Transformer is used instead of a TransformerXL, (2) latent state and actions tokens are fused instead of having separate tokens.
    - Currently, it is not clear why STORM achieves better results than TWM. Is it due to implementation details or the proposed modifications actually matter? And if they matter, why?

- The results have limited significance. Although STORM outperforms TWM, it only yields marginal improvements over DreamerV3, which was not specifically optimized for Atari100k. Moreover, recent model-free methods [1, 2] achieve similar or better results in the benchmark.

- Ablations and additional experiments fail to provide additional insights:
    - lines 237-238: IRIS also relies on an autoencoder trained with a reconstruction loss and it still obtains good results on Breakout and Pong.
    - l261: comparing the number of layers for STORM and IRIS/TWM is not straightforward since the hidden dimension is not the same (256 vs 512).
    - 5.2: it is hard to draw any conclusions as only a few environments were considered and we do not know how they were picked. Also it is not clear if including z_t yields statistically signficant improvements.
    - 5.3: the premise of this section is interesting but again too few environments are considered and results have limited statistical significance. The improvement on Freeway over previous methods relies on the addition of an extra demonstration while the other methods use exploration strategies that do not involve expert data. It would be interesting to know whether STORM leverages demonstrations better than other world models.

**Questions:**

### Do you use sticky actions?

---

I ran your code, on the 4 following games, with and without sticky actions:



| Environment | Reported | Without sticky actions | With sticky actions |
| --- | --- | --- | --- |
| BankHeist | 641 | 1044.0 | 218.5 |
| Breakout | 16 | 29.3 | 11.2 |
| MsPacman | 2673 | 2942.0 | 1921 |
| PrivateEye | 7781 | 4458.4 | 100.0 |

Can you clarify whether you used the Atari environments with (`ALE/<envname>-v5`) or without (`<envname>NoFrameskip-v4`) sticky actions? It is not mentioned in your paper but the code seems to use the v5 environments by default (TWM, IRIS and DreamerV3 do *not* use sticky actions).

It seems that my `v5` results are significantly below the reported results, and that the `v4` are on-par/slightly better.


### Ablations to investigate the differences with TWM

---

The paper would greatly benefit from a thorough investigation of the differences with TWM. It should be made clear why STORM performs better than TWM as it seems like a variant with two minor modifications.

- Is the substantial performance improvement explained by implementation differences (in particular, there is [an open issue about reproducibility](https://github.com/jrobine/twm/issues/3) on TWM’s repo), or by the two modifications?
- If these modifications matter, can you run ablations to demonstrate their effectiveness?
- Maybe it is due to the incorporation of tricks for policy learning from DreamerV3?

I am keen to significantly increase my overall rating if this concern is properly addressed during the rebuttals.

### Other concerns

---

- In my opinion, there would be a substantial gain to experiment in more complex environments, e.g. Crafter [3], Minecraft [4], or Memory Maze [5]. Such results would expand our knowledge of what Transformer-based world models can achieve.

- Can you include the recent work on the benchmark [1, 2] in the related work section?

---

### References

- [1] *Sample-Efficient Reinforcement Learning by Breaking the Replay Ratio Barrier*. D'Oro, Pierluca and Schwarzer, Max and Nikishin, Evgenii and Bacon, Pierre-Luc and Bellemare, Marc G and Courville, Aaron. The Eleventh International Conference on Learning Representations, 2022.
- [2] *Bigger, Better, Faster: Human-level Atari with human-level efficiency*. Schwarzer, Max and Obando-Ceron, Johan and Courville, Aaron and Bellemare, Marc and Agarwal, Rishabh and Castro, Pablo Samuel. arXiv preprint arXiv:2305.19452, 2023.
- [3] *Benchmarking the Spectrum of Agent Capabilities*. Hafner, Danijar. International Conference on Learning Representations, 2021.
- [4] *Minerl diamond 2021 competition: Overview, results, and lessons learned*. Kanervisto, Anssi and Milani, Stephanie and Ramanauskas, Karolis and others. NeurIPS 2021 Competitions and Demonstrations Track, 2022.
- [5] *Evaluating Long-Term Memory in 3D Mazes*. Pasukonis, Jurgis and Lillicrap, Timothy P and Hafner, Danijar. The Eleventh International Conference on Learning Representations, 2022.


**Limitations:**

The authors discussed some technical limitations in the conclusion.

---

> ### Author Rebuttal · Authors · 2023-08-06
>
> Thanks for your effort spent reviewing our paper and providing many valuable suggestions. We will include the suggestions in the revised version. Below, we want to address the main concerns raised in the review.
>
> - Response to **Weakness 1** and **Question 2** about `comparison with TWM`: For further experiments, please refer to our **Response 3** in the **Author Rebuttal** above. As you pointed out, STORM demonstrates a training speed twice as fast as that of TWM, while also achieving superior performance scores. We believe that this achievement represents a significant and meaningful innovation.
>
> - Response to **Weakness 2** `marginal improvements compared to DreamerV3 & not as good as BBF[2]`: Although the improvement in overall score was not significant (112% to 126%), STORM is equal or better than DreamerV3 on 21 out of 26 games.
>
>   While global human mean and median metrics offer an overall view of an RL algorithm's performance, it is common to observe variations in different environments and methods. To illustrate, let's consider BBF[2] for comparison:
>
>   | Algorithm | Human mean & human median | No. of envs that STORM $\approx$ BBF | No. of envs that STORM > BBF |
>   | :-------: | :----------: | :-------------: | :-----------: |
>   |   STORM   |   1.267 & 0.584  |    4   |   5   |
>   |    BBF    |     2.247 & 0.917    |     -    |   -   |
>
>   Although BBF achieves nearly double the overall score compared to STORM, our method still outperforms it in five games, namely *Gopher*, *Hero*, *MsPacman*, *PrivateEye*, and *Qbert*. *Hero* and *PrivateEye* are two representative games in the Atari 100k benchmark that involve long-term exploration without rewards. In cases where BBF surpasses STORM, there are several environments where RL agents typically perform much better than humans. We believe that conducting research in such environments does not align with the expectations for future RL algorithms.
>
> - Response to **Weakness 3**: We mainly choose factors that may have a major impact on the performance of STORM for ablation study. Previous works like DreamerV3, IRIS, or TWM have also conducted ablation studies on their distinct contributions, such as policy training strategies, number of tokens, policy input selection, and other hyperparameters. Let us clarify your concerns:
>
>   - The ablation study on the number of layers is not meant for comparison with other methods, but to provide readers with insights into STORM's configuration. In the realm of deep learning, increasing the number of layers in neural networks may generally lead to improved performance, especially with residual connections. However, such a correlation is not evident in STORM. As discussed in **Sections 5.1 and 6**, we hope this analysis can inspire researchers interested in designing new model-based RL algorithms or exploring novel environments.
>   - Conducting ablation studies on the entire benchmark demands approximately **7 (ablation studies) $\times$ 0.2 (days per game) $\times$ 5 (seeds) $\times$ 26 (games) = 182  (NVIDIA GeForce RTX 3090 days)** ,which is resource-intensive and infeasible effort given our time and funding budget at present. Therefore, we have chosen representative environments that are sensitive to different configurations, based on our experience with STORM's development. Of course, users should customize the configuration of STORM or the selection of the algorithms according to specific environments when employing as discussed above.
>   - For the demonstration trajectories, please refer to our **Response 1** in the **Author Rebuttal** above.
>
> - Response to **Question 1** about `the use of sticky actions`:
>
>   - It's correct that we use `v5` environments as described in `train.py` and `train.sh`. However, we will respectfully disagree with the result that you provided. We re-conducted experiments on your listed environments (`v5`) and obtained similar results to what we reported in the paper. We suspect that the discrepancy might arise from the single seed used for training in your case, whereas the reported results in the paper represent the average of 5 runs (retrained with different seeds). As plotted in **Appendix A**, all results in your table fall within a reasonable error range. If all 5 seeds produce such results, please kindly let us know, as it would be quite surprising.
>   - The use of sticky actions, recommended in [1], allows us to verify the algorithm's robustness. At the project's outset, we decided to adopt the latest environment configuration, using `v5` instead of `v4`.
>   - We have also tested our algorithm on your listed environments without sticky action and found that, while there was no significant difference in the other three environments, there is an improvement for the results on the game *Breakout*. This improvement can be attributed to the sensitivity of actions like catching, serving the ball, and moving the slider in this type of game.
>
> - Response to  **Question 3**, other concerns:
>   - Thanks for your helpful suggestion. Agreed. Indeed, performing additional experiments on new/complex environments would be helpful for sufficiently demonstrating the efficiency of STORM and understanding the performance limit of Transformer-based world models for RL. Nonetheless given our time and funding budget, we will leave it for future research. Please also refer to our **Response 1** in the **Author Rebuttal**.
>   - Thank you for your suggestion, and we will include these two papers in the related work section.
>
>
> ---
>
> **References**
>
> [1] Machado, Marlos C., et al. "Revisiting the arcade learning environment: Evaluation protocols and open problems for general agents." *Journal of Artificial Intelligence Research* 61 (2018): 523-562.
>
> [2] *Bigger, Better, Faster: Human-level Atari with human-level efficiency*. Schwarzer, Max and Obando-Ceron, Johan and Courville, Aaron and Bellemare, Marc and Agarwal, Rishabh and Castro, Pablo Samuel. arXiv preprint arXiv:2305.19452, 2023.

---

> > ### Comment · Reviewer_rfCq · 2023-08-14
> > **Thanks for the rebuttal!**
> >
> > I thank the authors for their response. I appreciate the effort made to compare STORM with TWM and the inclusion of missing papers in the related work.
> >
> > Overall, the idea of the method suffers from a lack of novelty compared to TWM. However, given that the proposed method is faster and that the results are significantly better than TWM (and obtained in environments with sticky actions), I am convinced that this work would be of interest to the community. I am updating my rating from 5 to 7.

---

> > > ### Author Response · Authors · 2023-08-15
> > >
> > > We appreciate you taking the time to read our rebuttal and reconsider our work. Thanks for your thoughtful feedback and recognizing our contributions.

---

### Official Review · Reviewer_4iYJ · 2023-07-06

**Soundness:** 3 good
**Presentation:** 3 good
**Contribution:** 3 good
**Rating:** 6
**Confidence:** 4

**Summary:**

The authors propose several modifications to the recently proposed Transformer based world model for Model-based reinforcement learning. Specifically, they come up with a single latent stochastic state and treat action as an explicit input to the state as opposed to a token (as in previous works) and show significant performance and speed in limited regime of Atari 100k tasks.

**Strengths:**

1. Improved performance as well as significant time reduction in STORM's agent training which is crucial in MBRL agents alongside sample efficiency.
2. Well written paper overall and was easy for me to read in a single go (NOTE: I'm very familiar with this MBRL space so that is another reason that has contributed to this but overall the flow of the paper was very intuitive.)

**Weaknesses:**

1. **Highlighting differences between TWM, Dreamerv3, Transdreamer, IRIS, SimPLe**:

	(a) It would be much better to have a table indicating different aspects such as (a) RNN (b) multiple tokens (c) Transformer variant etc. This would be much easier for the reader to read. Additionally, I did not find the motivation behind some of these modeling choices (see below for detailed descriptions).

	(b) On L127 the authors write "In contrast to IRIS [12] that employs multiple tokens, STORM utilizes a single stochastic latent variable to represent an image." This is correct but TWM encoding style and STORM's encoding style are the same -- so I find it a bit misleading to omit TWM as all three (TWM, IRIS and STORM) are transformer models

	(c) On L129, re: "STORM follows a vanilla Transformer [15] structure, while TWM [11] adopts a Transformer- XL [19] structure." -- it would be helpful why one would like to use a vanilla Transformer as opposed to Transformer-XL? What are the benefits?

	(d) On L131, "TWM [11] treats observation, action, and reward as three separate tokens of equal importance." -- I am not sure if this statement is true. The input to the transformer in the case of TWM is (obs, action, rew) but that doesn't imply that they are equal -- the attention weights of the transformer would be the ones deciding whether to consider these tokens or not. See Figure 6 of TWM paper showing the attention map over $(s, a, r)$ and it is clear that not all are weighted equally.

	(e) On L134, "Unlike Dreamer [8] and TransDreamer [16], which incorporate hidden states, STORM reconstructs the original image without utilizing this information." -- What is the additional benefit of not using the hidden state? I don't think there is a significant reduction in time for the reconstruction.

2. **Performance**:
	(a) What specific component according to the authors attributes to the superior performace of STORM. L235-237 mention the self-attention mechanism -- which is a valid argument against the RNN based methods. However, it is unclear to me what is helpful in STORM when compared to IRIS or TWM.

	(b) L237-238: what do authors mean by the "nature of autoencoders?" For example, the encoding style in TWM and STORM is very similar. It is unclear to me why specifically to single object games does STORM perform poorly. I'd like the authors to elaborate on this.


3. **"Decoder at rear" experiment**: I am not sure what exactly the purpose of this experiment was. From what I understand, the model can be formulated as TSSM (as mentioned in TransDreamer paper), and hence the the reconstruction would be using the posterior $z_t$ and not the prior $\hat{z}_t$. Is there something that I'm missing here?

4. **Impact of trajectory**:
    (a) How exactly was the trajectory used in the replay buffer? Was there any prioritized replay buffer or was the sampling of trajectories for the training of world model uniform?
    (b) Inclusion of trajectory is *not* specific to STORM -- so I'm not sure how this experiment validates the usefulness of STORM specifically and not apply to other world models (TWM, IRIS etc).


**[Very minor comment -- not considered for rating the paper]**

5. In appendix, Table 9, it would be more clearer to denote the "Imagination batch size" by another symbol or $B \times T$ as that is effectively what happens during imagination.


----
**Rationale for my rating**
I think the model details mentioned in the paper are important to share with the broader community, however I do think that motivation for several choices made in the modeling of STORM that has *not* been explicitly provided.

Additional it is not clear to me that experiments such as "Decoder at rear" or the inclusion of trajectory are fundamentally because of STORM and seem more of a study on Dreamer-*like* paradigm.

I would like the authors to address my concerns above and for the time being, I have leaned towards Borderline reject. I will update my rating post-rebuttal and discussion with the authors.

-----
**Post-rebuttal rating**

Based on the rebuttal provided by the authors, I've decided to bump up my rating to *Weak Accept* as they have addressed a majority of my concerns. The only section that I'm currently unconvinced is the inclusion of demonstrations which is not the major contribution of this work and would require its own analysis instead (a single section won't suffice in my opinion). I once again thank the authors for their comprehensive rebuttal.

**Questions:**

See **Weaknesses** section.

---

> ### Author Rebuttal · Authors · 2023-08-06
>
> Thanks for your effort spent reviewing our paper and providing many valuable suggestions. We will include the suggestions in the revised version. Below, we want to address the main concerns raised in the review.
>
> - Response to **(1a, 1b)**: We sincerely appreciate your suggestion, and we will include a table of differences between these algorithms in the updated version to enhance the clarity and motivation of the modeling choices.
>
> - Response to **(1c)**: Transformer-XL was proposed to increase attention length and handle extremely long text in the NLP tasks. However, a vanilla Transformer structure like BERT was still proved to have a memory capacity of 512 tokens (as reported in their paper, Appendix A.2), which is much longer than the regular sequence modeling horizon in RL problems. STORM, Dreamer, IRIS, and TWM typically have a sequence modeling horizon ranging from 16 to 64. Given these considerations, the use of Transformer-XL is unnecessary and may lead to a drawback in performance and runtime.
>
> - Response to **(1d)**: We apologize for this confusion. As correctly pointed out,  TWM does provide different attention weights to different tokens. What we want to express here is that all observations, actions, and rewards are involved in the same self-attention process as input tokens, and performing self-attention across different types of data (observation, action, and reward of different physical meanings) may potentially negatively impact or limit the performance (**L112**).
>
> - Response to **(1e, 3)**, `Why ablations on "decoder at rear"`: Yes you are correct about this, "decoder at rear" means the the model is formulated as TSSM.
>
>   DreamerV3 is an RNN-based algorithm, that enables the fusion of observations and actions into the hidden state step by step, which naturally facilitates the observation reconstruction with historical information. In contrast, Transformer-based world models process a sequence of observations and actions simultaneously, leading to two options for the decoder position: **(1) decoder at rear** (like TSSM in TransDreamer) to reconstruct the original observation using the hidden state $h_t$ and prior $\hat{z}_t$, and **(2) decoder at front** (STORM), which resembles the original structure of the variational autoencoder.
>
>   TransDreamer needs a great number of samples to converge in Atari games (see Figure 5 in their paper), which inspired us to further investigate it. As the ablation study presented in **Section 5.1**, we found that the structure of STORM is superior, and the "decoder at rear" approach struggles to converge with a 100k sample budget. This suggests that the variational reconstruction loss may not effectively drive the training of a Transformer-based model between the stochastic variable and the decoder, or the reconstruction loss might be excessively influenced by the KLDiv loss. In conclusion, this design is primarily for improving final performance, rather than lowering computational complexity.
>
> - Response to **(2a, 2b)**: Please refer to our **Response 2&3** in the **Author Rebuttal** above.
>
> - Response to **(4a)**: In our experiments, the samples in the demonstration buffer and the online buffer are uniformly sampled separately. For initiating the imagination process, a batch consists of 4 demonstration trajectories and 16 online trajectories.
>
> - Response to **(4b)**: Please refer to our **Response 4** in the **Author Rebuttal**.
>
> - Response to **(5)**: We appreciate your constructive suggestion, and we will explain the batches further in the header of Table 9 in the revised paper.

---

> > ### Comment · Reviewer_4iYJ · 2023-08-21
> > **Thank you for the rebuttal. Most of my concerns have been addressed!**
> >
> > Thanks to the authors for the detailed rebuttal and response to all my questions. I appreciate their effort on this end.
> >
> > **1c** I agree with the rationale and for a benchmark on ATARI more than 512 tokens in unnecessary. However, for more long-horizon tasks where memory plays a key role, Transformer-XL can be good choice (as shown in TransDreamer).
> >
> > **1d** Thanks for acknowledging this. I do agree that computationally it does get expensive to send an additional *n* tokens for rewards -- especially if we are thinking of real-world experiment or might lead to negative learning of policy.
> >
> > **3** Thanks for the detailed explanation on this and adding this to the main paper would be incredibly helpful to the reader.
> >
> > A majority of my concerns (1-3) are addressed however I am not fully on-board with the rationale behind Section 5.3 **Impact of the demonstration trajectory**. The benefits of inclusion of a demonstration is trivial and has been shown in [1]. However, as most of my concerns are addressed I'd like to increase my vote to Weak Accept and vote towards acceptance of the paper.
> >
> > Contrary to reviewer E5z3, I do believe that Atari 100k is a challenging and sufficient benchmark to evaluate this work. Additional experiments on other environments would be nice to have but the current experiments, in my opinion, are sufficient to back up the claims.
> >
> > ----
> > References:
> > [1] Multi-View Masked World Models for Visual Robotic Manipulation, Younggyo Seo, Junsu Kim et al, ICML 2023

---

> > > ### Author Response · Authors · 2023-08-21
> > >
> > > Thanks very much for taking the time to read our rebuttal and updating your score! Thanks for your thoughtful feedback and recognizing our contributions!
> > >
> > > We believe that the inclusion of `demonstration/expert trajectories` is full of potential and remains to be further explored. Indeed, our current investigation of this idea is via a toy example but it may lead to a possible solution to few-shot RL in the future. Thanks for pointing out the reference! This concurrent work also reveals the benefits of the technique (it was published on May 31st, while the NeurIPS abstract deadline is May 11th).
> > >
> > > We also thank you for supporting our experiment settings on Atari 100k.

---

> > > > ### Comment · Reviewer_4iYJ · 2023-08-21
> > > >
> > > > Sure I agree with your comment on the use of demonstrations as a toy example/proof-of-concept and the fact that MV-MWM was accepted very recently.
> > > >
> > > > However, the lack of comparison with Dreamer, TWM and other methods threw me off while reading that section as it had nothing *specifically* to do with STORM and applies to any general MBRL framework.
> > > >
> > > > That being said, I hope that this work further encourages researchers to investigate how/where/when do demonstrations lead to better performance in MBRL.

---

> ### Comment · Area_Chair_8ViK · 2023-08-17
>
> Dear Reviewer,
>
> The author has posted their rebuttal, but you have not yet posted your response. Please post your thoughts after reading the rebuttal and other reviews as soon as possible. All reviewers are requested to post this after-rebuttal-response.

---

### Official Review · Reviewer_Cg1v · 2023-07-07

**Soundness:** 3 good
**Presentation:** 3 good
**Contribution:** 3 good
**Rating:** 5
**Confidence:** 3

**Summary:**

The paper presented a Transformer-based model-based RL framework. As with earlier approaches, online data is gathered into the replay buffer with the learned reactive policy, the Transformer-based world model is trained with segments sampled from the replay buffer, then the policy is optimized with imaged data generated from the world model. The improved results were reported on the Atari 100K benchmark.  The authors also conduct a thorough ablation study to justify heir architectural design choices.

**Strengths:**

This paper is well-organized and easy to follow.

**Weaknesses:**

- What criteria were employed for task selection in the ablation studies?  They are not consistent over all the subsections, nor do they have the same trend in Table 1 compared to baseline models. The proposed method performs better in some tasks, while it doesn't in others. To provide a more robust evaluation of their proposed method, it would be better to consider extending the ablation studies to a wider range of tasks
- It would also strengthen the paper if the authors could provide the converged results on the full Atari benchmark.

**Questions:**

Please see the weaknesses.

**Limitations:**

yes.

---

> ### Author Rebuttal · Authors · 2023-08-06
>
> Thanks for your effort spent reviewing our paper and providing many valuable suggestions. We will include the suggestions in the revised version. Below, we want to address the main concerns raised in the review.
>
> - Response to **Weakness 1**`criteria for task selection in the ablation studies`:
>   - Conducting ablation studies on the entire benchmark demands approximately **7 (ablation studies) $\times$ 0.2 (days per game) $\times$ 5 (seeds) $\times$ 26 (games) = 182  (NVIDIA GeForce RTX 3090 days)**, which is resource-intensive and infeasible effort given our time and funding budget at present. Therefore, we have chosen representative environments that are sensitive to different configurations, based on our experience with STORM's development.
>   - Previous works like DreamerV3, IRIS, and TWM conduct ablation studies on their special contributions, such as policy training strategies, number of tokens, choice of policy Input, and other hyperparameters. In **Section 5.1**, we present the impact of model design and configuration, which are essential considerations when implementing a Transformer-based world model structure. In **Section 5.2**, we delve into the impact of policy input selection, an aspect also studied in TWM, providing researchers with valuable insights into STORM and model-based reinforcement learning. In **Section 5.3**, we analyze the influence of a single demonstration trajectory, highlighting that the challenging exploration issue in RL can potentially be addressed by integrating external knowledge and world models in future research. We believe this analysis could inspire further investigations in this field.
>   - Regarding concerns about performance (`The proposed method performs better in some tasks, while it doesn't in others`), we also address them in our response to **Reviewer rfCq Weakness 2**.
>
> - Response to **Weakness 2** `It would also strengthen the paper if the authors could provide the converged results on the full Atari benchmark`: Please refer to our **Response 1** in the **Author Rebuttal** above.

---

> > ### Comment · Reviewer_Cg1v · 2023-08-17
> > **Response**
> >
> > Thank the authors for the clarification. I would like to keep my score as it is.

---

> > > ### Author Response · Authors · 2023-08-18
> > >
> > > We express our gratitude for your consideration of our rebuttal. Please kindly let us know if you have further questions. Also please consider raising your score if you agree with our rebuttal.

---

### Author Rebuttal · Authors · 2023-08-06

We express our gratitude for the valuable feedback and for the recognition of STORM's efficiency as emphasized by reviewers (4iYJ, rfCq, E5z3), along with the commendation for the paper's coherent structure and presentation, as indicated by reviewers (Cg1v, 4iYJ, rfCq). In the subsequent discussion, we address the main common points raised in the reviews.

1. Response to `STORM should be extended to other tasks like MuJoCo, Crafter, Minecraft, etc`: We appreciate the constructive suggestion and acknowledge the potential benefits of extending STORM to real-life applications like robot control or real-time game control learning tasks. However, due to the substantial computational resources required, evaluating our algorithm on a wide range of complex benchmarks, such as MuJoCo, Crafter, and Minecraft, is currently infeasible given our time and funding budget. For instance, to evaluate our algorithm on full Atari games with 200M sample steps, it demands approximately **5 (days per game) $\times$ 5 (seeds) $\times$ 55 (games) = 1375 (NVIDIA GeForce RTX 3090 days)**. While we aim to explore such extensions in the future, we think it is beyond the scope of this paper.

   Meanwhile, the selection of the 26 Atari 100k games in the experiments is routine and has been done in previous work such as IRIS, DreamerV3, TWM, SimPLe, the baseline methods used in our paper.  It is thus commonly acknowledged that such a selection and evaluation is sufficient to verify the effectiveness of these algorithms. We believe such an evaluation method is sufficient to verify the effectiveness of the algorithm.

2. Response to `Why IRIS, a Transformer based method, performs better than STORM on some tasks`:  IRIS is different from SimPLe, Dreamer, TWM, and STORM. It maps an image observation to 4$\times$4 or more tokens through a VQ-VAE structure. This design is capable of more precisely capturing the position and motion of small objects like in *Pong* and *Breakout*. However, the advantage comes with a trade-off in terms of slow training and inference speed.

3. Response to `Extra ablation studies on structures of STORM and TWM`: We made modifications to our code so that the world model is organized as TWM, and did **addtional experiments** on several environments. We trained the world model with a vanilla Transformer and trained the agent's policy with $s_t = [h_t, z_t]$. We did not use their "balanced dataset sampling" trick. The results obtained are as follows:
   | Environment | STORM (paper) | STORM modified as TWM | TWM (paper) |
   | :---------: | :-----------: | :-------------------: | :---------: |
   |    Alien    |      984      |          822          |     675     |
   |  BankHeist  |      641      |          394          |     467     |
   |  Breakout   |      16       |          12           |     20      |
   |    Pong     |      11       |           7           |     19      |
   |  MsPacman   |     2673      |         1829          |    1588     |
   |   UpNDown   |     7985      |         7092          |    15982    |

   Using the TWM modeling approach in several environments can harm the performance. Still, we acknowledge that comparing algorithms with different implementations can make fair ablation studies challenging.

   However, it is crucial to highlight that even though STORM and TWM achieve similar scores, the training cost differs significantly due to the $\mathcal{O}(n^2)$ complexity of self-attention operation, with a ratio of $3n_{\mathrm{STORM}}=n_{\mathrm{TWM}}$. Specifically, the training hours required on a GeForce RTX 3090 are as follows:

   |   Algorithm    | STORM | STORM modified as TWM | TWM  | Real sampling time |
   | :------------: | :---: | :-----------------: | :--: | :----------------: |
   | Training hours |  4.3  |        10.5         | 12.5 |        1.85        |

   It is noteworthy that current model-based RL algorithms exhibit slower convergence rates compared to real-time sampling, which means one will need several expensive computing devices to control a DayDreamer[1]-like robot. In contrast, using STORM is more economical and environment-friendly under such circumstances.

4. Response to `Demonstration trajectories`: We agree that the inclusion of demonstration trajectories may also benefit similar algorithms like TWM, IRIS, or Dreamer. While developing our approach, we found that certain games, such as *Freeway*, appear relatively easy for humans who understand or infer the rules quickly from past experience, but they are challenging for algorithms like STORM, DreamerV3, and IRIS. Interestingly, we experimentally observed that including even a single demonstration trajectory has a substantial impact on the performance of *Freeway*, confirming similar findings mentioned in the Appendix H of the IRIS paper.

   We believe that combining demonstration and model-based RL offers a promising solution to address the challenging exploration issue. However, we acknowledge that this method has some limitations, as discussed in **Section 5.3 (Lines 289-292)**, possibly due to the neglect of on-policy distribution, as mentioned in **Section 6 (Lines 321-325)**. As a result, we present the results on three typical games to demonstrate the potential benefits of demonstration trajectories while acknowledging these limitations. We hope to draw the attention of other researchers to further explore this avenue and evaluate the merits of integrating demonstration trajectories with model-based RL techniques.

---

**References**

[1] Wu, Philipp, et al. "Daydreamer: World models for physical robot learning." *Conference on Robot Learning*. PMLR, 2023.

---

### Decision · Program_Chairs · 2023-09-21

**Decision:**

Accept (poster)

**Comment:**

The paper presents STORM, an architecture for model-based reinforcement learning that effectively combines Transformers and variational autoencoders. It demonstrates noteworthy performance improvements and time efficiency in Atari benchmarks, marking a valuable contribution to the field.

The manuscript is well-organized, technically sound, and offers meaningful advances in both performance and computational efficiency, as highlighted by the reviewers. However, there are suggestions for improvement, such as extending ablation studies, providing deeper motivations for modeling choices, and additional analysis on the use of stochastic variational autoencoders. Some of these concerns have been partially addressed in the rebuttal. Considering the overall quality of the work and the improvements made post-rebuttal, the paper is deemed worthy of an accept decision.